# How Mentally Fatiguing Are Consecutive World Padel Tour Matches?

**DOI:** 10.3390/ijerph18179059

**Published:** 2021-08-27

**Authors:** Jesús Díaz-García, Inmaculada González-Ponce, Miguel Ángel López-Gajardo, Jeroen Van Cutsem, Bart Roelands, Tomás García-Calvo

**Affiliations:** 1Faculty of Sport Sciences, University of Extremadura, 10004 Caceres, Spain; jdiaz@unex.es (J.D.-G.); mianlopezg@gmail.com (M.Á.L.-G.); tgarciac@unex.es (T.G.-C.); 2Faculty of Education, University of Extremadura, 06006 Badajoz, Spain; 3VIPER Research Unit, Royal Military Academy, 1000 Brussels, Belgium; jeroen.van.cutsem@vub.be; 4Human Physiology and Sports Physiotherapy Research Group, Vrije Universiteit Brussel, 1050 Brussels, Belgium; Bart.Roelands@vub.be; 5Sport and Exercise Science, School of Psychology and Life Sciences, Canterbury Christ Church University, Kent CT1 1QU, UK

**Keywords:** mental effort, motivation, reaction time, professional padel, racket sports

## Abstract

It is currently unknown whether mental fatigue occurs throughout a WPT competition and whether consecutive matches affect how mentally fatiguing a match is perceived to be. The objective was to quantify the effects of successive professional matches on mental fatigue. A total of 14 professional players (9 males, Mage = 25, 5 females, Mage = 21) participated during qualified rounds of a WPT with three eliminatory matches: Match 1 (morning) and 2 (afternoon) on day 1 (*n* = 14), Match 3 (morning) on day 2 (*n* = 6). Mental fatigue and motivation, with scales, and reaction time, with a 3-min Psychomotor Vigilance Test, were measured at two time intervals (pre and post matches (<30 min)). To analyze the evolution of these variables, a two-way repeated measures MANOVA was performed. An increase in mental fatigue from pre- to post-matches was observed (*p* < 0.01), with an accumulation of mental fatigue between matches played on day 1 (*p* < 0.01), maximizing the mental fatigue perceived during Match 2. Padel matches impair motivation and reaction time (*p* = 0.04), without effects between successive matches, which reinforced the idea that mental fatigue may impair padel performance (i.e., reaction time). Coaches should use training interventions and recovery strategies to counteract/avoid the accumulation of mental fatigue during professional tournaments.

## 1. Introduction

The professionalization of padel (e.g., the World Padel Tour (WPT)) has triggered the interest of multiple researchers in this sport. This interest, currently, has resulted in insights into the effects of padel matches on physical and technical–tactical fatigue [1,2]. These studies indicated that padel implies high-intensity efforts performed during a long time, although short periods of aerobic rest occur between these high-intensity efforts. Specifically, Llin-Mas et al. [3] described that the average heart rate in professional padel players during official matches was 151.70 ± 15.07 bpm. The authors further highlighted that the time that professional padel players spend over 160 bpm during play time is 29.6 min per game. Padel also implies a great variety of strokes during efforts that induce physical fatigue. Sañudo-Corrales et al. [4] described that, in high-level male padel players, the volley is the main stroke used by these players; meanwhile, drive and backhand with and without glass and smash were also widely used. To perform these strokes adequately, padel players should show a good previous displacement; Llin-Mas [5] analyzed the distance covered by point according to the player´s role. The authors indicated that in a sample of high-level players, the player that performed the service ran an average distance of 13.61 m per point, meanwhile their teammate covered less distance during the point with an average of 10.45 m per point. This information allows coaches to design training sessions according to the padel matches’ demands. Furthermore, it is known that padel imposes a high mental demand [6]; however, the potentially mentally fatiguing aspect of padel matches remains to be elucidated. 

Mental fatigue is a psychobiological state caused by prolonged mental effort (e.g., high cognitive load tasks, emotion control before starting a match) [7]. On the one hand, padel implies cognitive effort to resolve competition problems by selecting the most appropriate solutions at high speed in an unpredictable environment [6]. On the other hand, padel is also associated with emotional efforts. A padel player needs to try and maintain self-confidence, and to be able to cope with interdependence (i.e., your teammate´s performance depends on your own performance [8]). Previous studies also suggested that the recovery of this mental fatigue is not immediate [9,10]. Therefore, although to our knowledge no previous studies have analyzed these variables during padel competitions, WPT players likely experience mental fatigue during a WPT tournament, which implies playing one or more match(es) per day during successive days. 

A limited number of studies have looked into the effects of mental fatigue on racket sport performance, but the outcome is not straightforward. Kosack et al. [11] have not reported impairments in a badminton specific test after a mental exertion task, and Le Mansec et al. [12] have not reported negative effects of mental fatigue on reaction time, an important variable in this type of sport. On the other hand, Le Mansec et al. [13] reported a decrease in the speed of the ball in table tennis players after a mental exertion task, probably to maintain the accuracy of the shots in the presence of mental fatigue. Meanwhile, Van Cutsem et al. [14] reported impairments in badminton players’ reaction time after a mental exertion task. Regarding subjective feelings of performance, Russell et al. [15] reported poor subjective feelings of performance in the presence of mental fatigue, specifically slower reactions or demotivation. Motivation seems to be an interesting variable in the effects of mental fatigue on performance; high levels of motivation could decrease the negative effects of mental fatigue with an increase in the effort perception tolerance [16]. However, the demotivation associated to mental fatigue [15] indicates an overload of the neural mechanism involved in the maintenance of performance when mental fatigue persists for a long period of time [17]. Consequently, a negative influence of mental fatigue could be present during a WPT tournament.

Based on the negative effects that mental fatigue could cause on padel performance and the possibility that padel players experience mental fatigue during a WPT tournament, the analysis of mental fatigue during a WPT seems justified. Therefore, the main purpose of this study was to quantify the changes in mental fatigue, motivation, and reaction time during a professional padel tournament. Secondary objectives of this study were to quantify the effects of emotional variables, specifically, one’s own and teammates’ performance satisfaction.

We hypothesized that mental fatigue, motivation, and reaction time values would be impaired by playing a competitive match (Hypothesis 1). We also hypothesized that mental fatigue would gradually build up during the tournament (Hypothesis 2). Finally, we hypothesized that emotional efforts associated with poor performance satisfaction would increase the mental fatigue (Hypothesis 3).

## 2. Materials and Methods

### 2.1. Sample

Twenty-six professional players during the WPT Madrid Open 2020 started this study. Twelve of these players lost M1 and subsequently were excluded from further analysis (see Design). Eventually, fourteen WPT players (nine males, Mage = 25.56, SDage = 6.77, MWPTranking = 103 and five females, Mage = 21.70, SDage = 3.85, MWPTranking = 81) competed in two consecutive matches and were included in the present study. All included players had trained almost six days per week for the last two years.

### 2.2. Instruments 

Visual Analogue Scale (VAS). The 10-cm Visual Analogue Scale was used to quantify the mental fatigue reported by players. Subjects were asked to indicate the perceived level of mental fatigue placing a mark on the VAS 10-cm line. The left side of the scale indicated “not at all”, while the right side indicated “maximum”. This subjective measure of mental fatigue has been used in several studies that quantify mental fatigue in sport activities [8]. The internal consistency of this scale was acceptable (Cronbach´s alpha = 0.78) with adequate temporal stability (test-retest *r* = 0.91).

Motivation. Motivation was asked about both pre- and post-match (How motivated are you to play padel?). The range of response was in format of Likert´s Scale (0–10), where 0 was “nothing” and 10 was the “maximum level possible”. 

Psychomotor Vigilance Task (PVT). The 3-min version of the Psychomotor Vigilance Task was used to measure the reaction time [18]. In this test, a visual stimulus appears in the center of the screen. When players push the button, the reaction time was calculated with the difference between the appearance of the stimulus and pushing the button. If the button was pushed before the stimulus appeared, a “false start” message was displayed on screen. The mean reaction time throughout the 3 min was calculated. 

Self and Teammate´s performance satisfaction. Self-performance satisfaction (How satisfied do you feel with your performance in the last match?), and teammate’s performance satisfaction (How satisfied do you feel with your teammate’s performance in the last match?) were assessed after the match. The responses ranged from 0 (“nothing”) to 10 (“maximum level possible”). 

This type of sample and design allows little time to assess subjective feelings of motivation, self’s and teammate’s performance satisfaction. Previous studies have justified the use of single item questions in these cases [19]. The internal consistency of this scale was acceptable (Cronbach´s alpha = 0.74) with adequate temporal stability (test–retest *r* = 0.87).

### 2.3. Design

All participants were informed about the objectives of the study and signed a participation agreement according to the local Ethics Committee. 

The study was performed during the qualifying rounds of the WPT Madrid Open 2020. WPT competitions imply a system of eliminatory rounds; therefore, couples that lose a match their participation in the tournament are immediately finished. The qualifying rounds are composed by three matches. With the objective of analyzing the evolution of mental fatigue throughout successive matches, we included an exclusion criterion: players had to win their first match, so they played at least two matches, to be included in the present study. The first match (M1) was played on the morning of day 1 of the tournament, while the second match (M2) was played in the afternoon of this same day (mean average time between matches was from 5 to 6 h). A total of fourteen players won M1 and thus qualified for M2; therefore, the evolution of mental fatigue across successive padel matches played on the same day 1 was evaluated with the data of these fourteen players (*n* = 14; this was the 44% of the participants qualified to this round; no naps were allowed in the study to avoid the effects of naps on mental fatigue). Subsequently, six of these players also won M2, and thus played a third match. This third match (M3) was played on day 2 of the tournament in the morning. All matches were between 90 min and 128 min long, with a medium time of 105 min per match. Therefore, the evolution of mental fatigue across padel matches played on successive days was evaluated with the data of these six players (*n* = 6; this was the 38% of the participants qualified to this round). Two measures per time (pre- and post-match) were quantified. Pre- and post-match measurements included mental fatigue, motivation and reaction time, while performance satisfaction was only measured post-match. Researchers emphasized, during the tournament to the players, the importance of answering the questionnaires as close as possible to the start and finish of their matches (both in a maximum range of 30 min).

### 2.4. Data Analysis

Statistical tests were performed with SPSS 25.0 (IBM, New York, NY, USA). All data were expressed as the mean ± SD. The Shapiro–Wilk normality test was checked, showing normal distribution for the variables described. Sphericity was verified by Mauchly’s test; when the assumption of sphericity was not met, the significance of F ratios was adjusted with the Greenhouse–Geisser procedure. Additionally, the Levene’s test was used to evaluate the variance homogeneity. First, two-way repeated measures ANOVAs were performed with two factors with both levels, match (M1 and M2) × time (pre- and post-match), with the 14 players that played two matches the same day 1 (*n* = 14). This method was used in order to test the effect of successive padel matches played on the same day on mental fatigue, motivation and reaction time. Secondly, two way-repeated measures ANOVAs were performed with three factors, match (M1, M2 and M3) x time (pre- and post-match), with the 6 players that played three matches (*n* = 6). This method was used in order to test the effects of successive padel matches played on consecutive days on mental fatigue, motivation and reaction time. If a main effect for match was observed, pairwise comparisons were performed according to Bonferroni. Finally, a one-way repeated measures ANCOVA was performed for each match separately (M1 and M2, *n* = 14; and M3, *n* = 6) × two measures (pre- and post-match), using the player’s own and teammate performance satisfaction as covariates, to test the effects of emotional aspects on mental fatigue, motivation and reaction time. Statistical significance of all analysis was set at *p* < 0.05. In all these cases, partial eta squared (*η*_p_*^2^*) effect sizes were calculated. An effect of *η*_p_*^2^* ≥ 0.01 indicates a small, ≥0.059 a medium, and ≥0.138 a large effect [20]. 

## 3. Results

### 3.1. Mental Fatigue, Motivation and Reaction Time between Matches Played the Same Day

Table 1 shows the results from pre- to post-match values of mental fatigue, motivation and reaction time in M1 and M2 (*n* = 14), the two professional padel matches played on the same tournament day. Mental fatigue significantly increased after a padel match (the pairwise comparison showed a significant change from Pre-M1 to Post M1, *p* = 0.02 and from Pre-M2 to Post M2, *p* < 0.001) and, in addition, it was also significantly higher in M2 than in M1 (pairwise comparison showed a significant change from Pre-M1 to Pre-M2, from Pre-M1 to Post-M2, and from Post-M1 to Post-M2 where *p* < 0.001 and from Pre-M2 to Post-M1 where *p* = 0.01). Indeed, this higher increase presented in M2 than in M1 seems to be caused by an accumulation of mental fatigue by playing successive on matches the same day, as evidenced the presence of a significant interaction effect. Motivation and reaction time significantly decreased after a padel match, without significant changes in both variables from M1 to M2, nor an interaction effect. 

### 3.2. Mental Fatigue, Motivation and Reaction Time between Matches Played in Successive Days

Table 2 shows the results from pre- to post-match values of mental fatigue, motivation and reaction time in M1, M2 and M3 (*n* = 6), representing two professional padel matches played the same day 1 (M1 and M2) and a third match (M3) played the next day 2. No significant changes for mental fatigue, motivation and reaction time were observed by the inclusion of a third match played on a consecutive day. 

### 3.3. Influence of Performance Satisfactions on Mental Fatigue, Motivation and Reaction Time

No significant effect of the player’s own and their teammate’s performance satisfaction was observed on mental fatigue in M1 (F = 3.10; *p* = 0.09, *n* = 14), M2 (F = 0.91; *p* = 0.36, *n* = 14) and M3 (F = 0.43; *p* = 0.58, *n* = 6). Players with poor self and teammate’s performance satisfaction showed a significant decrease in motivation in M1 (F = 10.12; *p* < 0.01 *n* = 14) and M2 (F = 6.84; *p* = 0.04, *n* = 14) without significant effects in M3 (F = 0.17; *p* = 0.72, *n* = 6). Meanwhile, no significant effect of poor self and teammate’s performance satisfaction was observed for reaction time both in M1 (F = 0.19; *p* = 0.66, *n* = 14), M2 (F = 0.77; *p* = 0.49, *n* = 14) and M3 (F = 2.59; *p* = 0.25, *n* = 6). 

## 4. Discussion

The main purpose of this study was to quantify the evolution of mental fatigue, motivation and reaction time during a WPT competition. The most important findings of this study were that playing professional padel matches significantly increased the mental fatigue reported by players and impairs motivation and reaction time. Significantly higher feelings of mental fatigue were reported in the second game on the same day 1 (M2) from an accumulation of mental fatigue derived from the previous game of the day. Night rest was able to alleviate mental fatigue. Only motivation was affected by performance satisfaction, with motivation decreasing more in players with poor performance satisfaction. 

### 4.1. Mental Fatigue 

We hypothesized that playing a competitive match would increase mental fatigue (Hypothesis 1), with a residual effect on successive matches (Hypothesis 2). We showed that professional padel matches indeed cause a significant increase in mental fatigue. Therefore, we can confirm Hypothesis 1. Additionally, we showed higher values of mental fatigue in the second match played on the same day with a significant interaction effect. This indicates that, during a WPT tournament, a full recovery of mental fatigue does not occur between matches played on the same day. Next, we also found that this carry-over of mental fatigue from one match to the subsequent match was not observed between matches played on different days. Therefore, we can only partially confirm Hypothesis 2. Regarding the causes of mental fatigue during padel competitions, we hypothesized that emotional efforts caused by poor performance satisfactions would also increase mental fatigue (Hypothesis 3). However, self and teammate’s performance satisfaction did not influence mental fatigue. Therefore, we cannot confirm Hypothesis 3.

Van Cutsem et al. [7] defined that mental fatigue implies both cognitive and emotional efforts. The cognitive effort of the padel matches could sprout from the complex tactical decisions that players take to select the most appropriate solution [6]. Different to other racket sports, padel implies the use of side and back walls which increases the variety of the technical–tactical responses (see physical and technical–tactical demands in the introduction section) more than in other racket sports such as tennis or badminton, where higher levels of entropy in sports have been related with the highest increases in mental fatigue [8,21]. This could explain why mental fatigue did not significantly change the performance in certain studies such as that by Kosack et al. [11]. However, no previous studies have tested the effects of mental fatigue on padel. To our knowledge, no previous studies have quantified the mental fatigue caused by a real competition in racket sports. However, previous studies confirmed that performing cognitively fatiguing tasks can increase mental fatigue in professional racket sports players [11,12,13,14]. Indeed, Thompson et al. [22] and Smith et al. [23] highlighted the need to implement this type of ecological study, because this information allows coaches to know the true psychological demands during competitions and to understand how they can train these demands in simulated training contexts. Unexpectedly, the results of the present study show that, between teammates, (e.g., feelings of anger) emotional processes did not significantly increase mental fatigue. Therefore, the increase in mental fatigue induced by playing a padel match seems to be mainly related to the tactical and cognitive requirements of padel matches, or individual emotional processes (e.g., anxiety or feelings of frustration). 

Regarding the between match recovery for mental fatigue, previous studies suggested that mental fatigue remains elevated for a period of time, although the cognitively fatiguing task had finished [9,10]. For example, the fact that the increase in the mental fatigue 24 h after a soccer match had finished was described in two different studies [22,24]. Although this is not clear in padel yet, and other previous papers have not reported negative effects of mental fatigue [11,12], the information provided in previous studies indicate that a decrease in padel performance may occur in presence of mental fatigue [13,14]. This situation highlights the need to test training or prevention strategies for mental fatigue, as indicated Van Cutsem and Marcora [25]. A previous study has suggested that training could improve the mental fatigue resistance during sport activities [26]. Prevention strategies like creatine or caffeine have previously been shown to positively affect mental fatigue perceptions [27,28]. These ergogenic strategies may be related to the accumulation of the extracellular adenosine at the brain, one of the main theories to explain the negative effects that mental fatigue causes on sport performance. The need to use this prevention strategies between matches played at consecutive but different day also has not been proved, where the higher time of rest or the influence of sleep [29,30] could explain the absence of accumulation for mental fatigue during successive days. This information should consider by coaches to avoid the accumulation of mental fatigue between matches.

### 4.2. Motivation

We hypothesized that motivation decreases after a padel match (Hypothesis 1). Our results indeed showed a decrease in motivation from pre- to post-match in the majority of the cases. Therefore, we can confirm the Hypothesis 1. 

Russell et al. [15] reported that decrements in performance are an important symptom of mental fatigue. It seems that high levels of motivation during sport activities enhance the participation of the facilitation system, a neural mechanism that increases the effort perception tolerance, allowing to better maintain performance [17]. However, the presence of fatigue (mental fatigue, mainly) for a prolonged period of time impairs this mechanism, evoking mental fatigue and, also, significantly decreasing motivation [17]. Less is known about the evolution of this mechanism between successive matches or days [31]. Therefore, we suggest that the mental fatigue induced during a padel match might decrease motivation. The relation between mental fatigue and motivation could also explain why after the second game the six players that won M2 (Table 2) showed higher values of motivation and showed only small increases from pre- to post-match values of mental fatigue. Players in the WPT qualifying rounds do not receive monetary awards. They have to win a minimum of five matches (three in the first qualifying round, and two in the second qualifying round) to obtain monetary awards. However, WPT refunds the hotel costs to players that win their last match of the day (they have to play the next day; *n* = 6 in this study). 

### 4.3. Reaction Time

Finally, we hypothesized that impairments in reaction time would occur after a padel match (Hypothesis 1). The two-way repeated measures ANOVA showed significant impairments in reaction time from pre- to post-padel matches. Therefore, we can confirm Hypothesis 1. Le Mansec et al. [12] explained that reaction time has two main parts: a central or cognitive part (first: attention to detect the stimulus) and a peripheral or neuromuscular part (second: to obtain faster physical responses). In this same study, Le Mansec et al. [12] did not report impairments in reaction time after a physical or mental (both conditions were tested) exertion task. On the contrary, our results suggest that mental and physical fatigue, induced by playing padel matches, impair reaction time. The impairment in reaction time caused by mental fatigue in racket sport players has been previously reported [14]. O’Keeffe et al. [32] reported that dual tasks (like padel) induced higher levels of arousal and fatigue and, as a consequence, more negative effects attributed to fatigue. 

We should be careful with the data obtained (for example, only two players registered in the M3 indicate a high probability of committing the type 2 statistical error). Due to the low number of the total sample, however, we would also like to highlight the difficulty of obtaining ecological data in these top-quality professional padel players. These data can allow professional padel coaches and players to create a better management of mental fatigue during World Padel Tour competitions. This is the first paper that analyzed mental fatigue during a professional padel competition. According to these results, playing two consecutive padel matches could increase mental fatigue before Match 2, and mental fatigue may cause a decrease in performance. Therefore, professional padel players should use nutritional (i.e., caffeine) or psychological strategies between matches to improve the recovery of the mental fatigue between matches. In future studies, more information seems necessary to improve the knowledge about this topic. For example, we should study the influence of the result on mental fatigue or create a new questionnaire to obtain specifical data during the participation in the competitions.

## 5. Conclusions

Mental fatigue significantly increased after a professional padel match. It seems that cognitive load or individual emotional processes (e.g., anxiety) are the main cause of this phenomenon, with a smaller influence on performance satisfaction. Playing a second padel match on the same day is related to experiencing more mental fatigue, which indicates the recovery time between matches played on the same day is insufficient to fully recover in terms of mental fatigue. This residual effect did not appear between matches played on successive days. Motivation may decrease after a padel match (we have to take into account the result and the order of the match), where negative emotions caused by poor performance satisfaction enhanced these decreases in motivation. Reaction time was also impaired after a padel match. This is presumably caused by both the physical and mental fatigue induced during the game.

## Figures and Tables

**Table 1 ijerph-18-09059-t001:** Mental fatigue, motivation and reaction time from pre- to post-two matches evolution (*n* = 14).

Variables		M1		M2		Evolution		
		Pre-	Post-	Pre-	Post-	Pre–Post (*η*_p_^2^)	Matches (*η*_p_^2^)	Interaction (*η*_p_^2^)
Mental fatigue	M ± SD	3.23 ± 1.01	3.69 ± 1.70	4.15 ± 1.63	5.85 ± 1.68	*F* = 10.69 *p* < 0.01 * (0.47)	*F* = 9.91 *p* < 0.01 * (0.45)	*F* = 2.95 *p* < 0.05 * (0.17)
Motivation	M ± SD	9.23 ± 0.83	7.77 ± 2.46	8.77 ± 1.70	6.69 ± 3.30	*F* = 7.84 *p* < 0.02 * (0.39)	*F* = 1.99 *p* = 0.18	*F* = 0.25 *p* = 0.63
Reaction Time	M ± SD	0.37 ± 0.10	0.39 ± 0.10	0.37 ± 0.09	0.42 ± 0.13	*F* = 5.25 *p* < 0.04 * (0.34)	*F* = 1.60 *p* = 0.23	*F* = 2.79 *p* = 0.12

Notes. * = *p* < 0.05.

**Table 2 ijerph-18-09059-t002:** Mental fatigue, motivation and reaction time from pre- to post-three match evolution (*n* = 6).

Variables		M1		M2		M3		Evolution		
		Pre-	Post-	Pre-	Post-	Pre-	Post-	Pre–Post	Matches	Interaction
Mental fatigue	M ± SD	3.60 ± 1.34	4.40 ± 1.95	3.80 ± 1.30	4.80 ± 1.64	3.60 ± 1.30	6.60 ± 2.61	*F =* 3.18 *p* = 0.15	*F =* 1.24 *p* = 0.34	*F =* 1.56 *p* = 0.28
Motivation	M ± SD	9.20 ± 0.83	7.60 ± 2.88	8.60 ± 1.52	9.40 ± 0.55	9.20 ± 0.84	8.00 ± 0.70	*F = 0*.93 *p* = 0.39	*F = 0*.71 *p* = 0.48	*F =* 2.74 *p* = 0.16
Reaction Time	M ± SD	0.38 ± 0.11	0.40 ± 0.11	0.38 ± 0.11	0.41 ± 0.18	0.37 ± 0.13	0.42 ± 0.17	*F =* 1.75 *p* = 0.26	*F = 0*.04 *p* = 0.94	*F = 0*.26 *p* = 0.64

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
