# Peer review of "How Mentally Fatiguing Are Consecutive World Padel Tour Matches?"

_ijerph, 2021, doi:10.3390/ijerph18179059_

Round 1

Reviewer 1 Report

Thank you for the opportunity to review the interesting manuscript "How mentally fatiguing are consecutive World Padel Tour-2 matches"
Fatigue in sports is an important issue. From the point of view of sports performance but also the point of view of sports injuries. I appreciate the authors' interest in sports, to which there is not enough space devoted to sports science. I see the lack of the study in the relatively low number of the total sample and the overall methodological design. A more interesting form of research would be the creation of a new standardized questionnaire and the association with participation during the tournament. The overall scientific contribution is small.

1. Some references to studies in the introduction belonging to the discussion (Kosack et al., Le Mansec et al. Van Cutsem et al.)
2. The hypothesis in the introduction does not have to be based on studies based on Cardenas et al. [5], 73 Courel-Ibañez et al. [3], and Russell et al. [12].
3. Any evidence of the validity of the visual analog scale? missing source
4. The same situations in "Likert´s Scale"
5. It would be interesting to compare mental fatigue after a winning match and in the evening

Author Response

Thanks in advance for your time to improve our manuscript.

Reviewer 2 Report

Thank you for the opportunity to review this manuscript.

Why padal was selected can be written more clearly. Does this focus more on padel industry or mental fatigue in general? More discussion on meetal fatigue is appropriated. 

There is no theoretical framework in the introduction and this piece can be added and how this study contributes to a particular theory.

day 1,2, can be replaced with Day 1, or Day 2. Easy to read.

Sample section should come first as Participants. Design section should be after Instruments

Author Response

Thanks in advance for your time to improve our work.

Reviewer 3 Report

First of all, I would like to congratulate the authors for the proposed article, since it is a novel topic in paddle tennis and with a high impact in terms of psychological training.

On the other hand, I would like to indicate some indications to try to improve the article and some questions that I would like the authors to answer

INTRODUCTION:

The information they put in is really relevant. However, when talking about mental fatigue, it would also be convenient to indicate the various studies that have quantified the technical-tactical effort (the tactical part affects decision-making as well indicated by the authors), and it would also be convenient to indicate the studies of the physical effort of players. male and female paddle tennis professionals, even in chair paddle tennis. This will help the discussion to relate physical, technical-tactical and mental fatigue. (Mehta, R. K., & Parasuraman, R. (2014). Effects of mental fatigue on the development of physical fatigue: a neuroergonomic approach. Human factors, 56 (4), 645-656.

 METHOD:

 it is very well structured and explained, but it is necessary to indicate some information that I consider really relevant:

  • ¿What was the average rest time between one match and the next? Rest time can be a key element in mental recovery.
  • Did the players nap during the break time? If they slept, indicate how long did they sleep? Sleep time can affect recovery from mental fatigue and reaction time (Åkerstedt, T., Knutsson, A., Westerholm, P., Theorell, T., Alfredsson, L., & Kecklund, G. (2004 Mental fatigue, work and sleep. Journal of psychosomatic Research, 57 (5), 427-433.). If the authors do not know the answers to these questions, indicate this information as study limitations
  • What was the average duration of the M1, M2 and M3 matches? The longer the game lasts, it may be that there is greater mental fatigue between the pre and post
  • In the statistical tests performed, it is necessary to indicate the statistics used to calculate the effect size and their values ​​for small, medium or large effects (ie: partial eta squared). This should be indicated in the data analysis section.
  • Also in the data analysis section, indicate that the pairwise comparisons were made with significance adjustment according to Bonferroni.
  • One of the key elements when passing a scale or questionnaire is its validity. Authors should indicate in which study the scales used were validated.

RESULTS

The results section is written very clearly and in a very understandable way. To improve it, it is proposed:

  • Indicate the effect size
  • The results of table 1 indicate a significant effect of the interaction of the variables match and measure. However, then pairwise comparisons are not indicated (ie: are there significant differences between the pre measure in match1 and the pre measure in match 2? Authors should indicate pairwise comparisons
  • In table 2, the main effect of both the match and the measure disappears. This is due to the what I consider the main weakness of the article, which is the sample size. Only 2 players registered in the M3 indicate a high probability of committing the type 2 statistical error. This should be commented on in the limitations of the study.

The discussion is very well structured in terms of the variables analyzed and the hypotheses raised. However, it is proposed to improve it:

  • Compare the fatigue of this study with the fatigue of other team sports, and especially with racket sports, relating it to aspects of physical and technical-tactical load between sports.
  • On line 229 the authors conclude that motivation decreased from pre to post. However, this does not coincide with the results of table 2, at the significance level, nor with the descriptive results of M2. Can the authors try to justify this?
  • The discussion suffers from a section of study limitations. Authors should indicate them.

CONCLUSIONS:

They are adequate and adjusted to the objectives. They will have to be reviewed in relation to the motivational results in Table 2.

Author Response

(The authors gave the same response as above.)

Round 2

Reviewer 1 Report

I recommend publishing after incorporating the suggestions of the reviewer 3

Author Response

Thanks for your time to improve our work!

Reviewer 2 Report

Thank you for the opportunity to review. 

Author Response

Thanks for your time to improve the quality of this manuscript

Reviewer 3 Report

Although I have indicated a major revision, the changes and comments that I suggest you make are very few, but very important. I am convinced that with the proposed changes (in blue print) you will have a very good article for this journal

Author Response

Thanks for your time to improve our manuscript
